# Profilin Pfy1 is critical for cell wall integrity and virulence in *Candida albicans*

Xun Sun,[1,2,3,4] Yueqing Wang,[1,3,4] Xiaomin Yang,[1] Xi Xiang,[3,4] Lili Zou,[3,4] Xiaowen Liu,[3,4] Gang Luo,[5] Qi Han[1]

**ABSTRACT** Profilin is a small actin-binding protein that plays an important role in actin polymerization. However, its functions in *Candida albicans*, the most prevalent fungal pathogen, remain unclear. Here, we report that profilin plays a crucial role in *C. albicans* morphogenesis and virulence. Deletion of profilin results in abnormal morphogenesis and impaired hyphal development. Furthermore, *pfy1Δ/Δ* is hypersensitive to cell wall stress and displays thicker cell wall than wild-type cells, indicative of a critical function of Pfy1 in cell wall integrity. In addition, our findings demonstrate that profilin is required for the virulence of *C. albicans* in a murine model of systemic infection. In conclusion, our work provides a promising target for developing antifungal drugs.

**IMPORTANCE** Our research revealed Pfy1 is not only involved in hyphal development but also essential for pseudohyphal formation in response to DNA damage agents methyl methanesulfonate (MMS) and $H_2O_2$. The disruption of *PFY1* resulted in striking morphological defects in both yeast and hyphal forms. Further investigation suggested that profilin plays a role in polarized growth of *Candida albicans* via binding with Act1, and contributes to cell wall remodeling. Both hyphal growth and cell wall integrity are the important virulence factors of *C. albicans*. Thus, *pfy1Δ/Δ* strains significantly reduced mortality rates in mice. These findings suggested that profilin could serve as a target for developing new antifungal drugs possibly for use in combination therapies with caspofungin, for treating invasive candidiasis.

**KEYWORDS** profilin, *Candida albicans*, cell wall, virulence

F ungal diseases are a global public health threat, causing more than one billion human infections and nearly two million deaths annually (1). In October 2022, the World Health Organization (WHO) published the first-ever fungal priority pathogen list, identifying 19 fungi with the greatest public health impact and emerging antifungal resistance risk. These pathogens were categorized into three priority groups (Critical, High, and Medium), and *Candida* species were designated as the highest priority, "Critical" (2). As listed in the Critical Priority Group, *Candida albicans* is the most common opportunistic pathogen in humans, which colonizes mucosal surfaces and gastrointestinal tract of healthy individuals; however, in immunocompromised patients, *C. albicans* can invade solid organs such as kidneys, liver, and spleen, causing life-threatening infection with high mortality despite antifungal treatment (3–5). The limited effectiveness of current antifungal drugs and the rising prevalence of antifungal resistance indicate a need to develop new therapies for invasive candidiasis (6).

Profilin is a ubiquitously expressed protein and serves as a key regulator of actin polymerization, playing a critical role in cellular function (7). In budding yeast, the *PFY1* gene encodes profilin, which has been identified as a multifunctional protein involved not only in regulating actin dynamics but also in maintaining $Ca^{2+}$ homeostasis, controlling the relative abundance of actin and tubulin, intracellular transport (8–

Address correspondence to Qi Han, hanqi@bucm.edu.cn.

Xun Sun and Yueqing Wang contributed equally to this article. Author order was determined alphabetically.

The authors declare no conflict of interest.

12). *Candida glabrata* Pfy1 is an ortholog of human profilin-4 (13). Although profilin is structurally conserved among eukaryotes, there is only limited sequence identity between yeast and mammals (9), suggesting that profilin could be a potential antifungal target. Indeed, Keigo Ueno et al. have designed novel peptides binding to the active interface of profilin against *Candida glabrata* (13), where *PFY1* is an essential gene. Unlike *Saccharomyces cerevisiae* and *Candida glabrata*, *C. albicans* does not require profilin for its survival (14); however, PEP-IA18, an antimicrobial peptide derived from the profilin of *Spodoptera frugiperda*, still displayed potent anti-fungal effects against *C. albicans* (15). Unfortunately, our understanding of the function and related mechanisms of profilin in *C. albicans* remains limited.

In this study, we deleted *PFY1* in *C. albicans* and conducted comprehensive phenotypic characterizations. Our findings demonstrate that Pfy1 plays a pivotal role in hyphal development, cell wall integrity, and virulence of *C. albicans*. Based on these results, we propose that targeting Pfy1 could be a promising therapeutic strategy for the treatment of invasive candidiasis.

## MATERIALS AND METHODS

### Strains and growth conditions

The *C. albicans* strains used in this study are listed in Table 1. *C. albicans* was routinely grown at 30°C with shaking at 200 rpm in YPD medium (1% yeast extract, 2% peptone, and 2% glucose). For growth on plates, 2% agar was added to the medium. To select for nourseothricin-resistant transformants, 200 µg/mL of nourseothricin (Werner Bioagents, Jena, Germany) was added to YPD agar plates (YPD-Nou plates). To obtain the nourseothricin-sensitive derivatives in which the *SAT1*-flipper was excised by FLP-mediated recombination, transformants were grown overnight in YCB–BSA medium (2.34% wt/vol yeast carbon base, 0.4% wt/vol bovine serum albumin, pH 4.0) to induce the *SAP4* promoter controlling *Ca*FLP expression, streak inoculated onto YPD plates containing 25 µg/mL nourseothricin, and incubated at 30°C for at least 2 d.

Hyphal growth was induced by supplementing YPD medium with 10% fetal calf serum and incubating at 37°C with shaking at 200 rpm or streaking yeast cells onto Spider agar plates (1% wt/vol beef extract, 1% wt/vol mannitol, 0.2% wt/vol $K_2HPO_4$, 2% wt/vol agar, pH 7.2) to incubate at 30°C for 7 d. Invasive growth was assayed using YPD plates grown at 30°C for 2 d. After this period, the plates were photographed before and after thoroughly washing the cells from the surface. Any cells remaining embedded in the agar were interpreted as evidence of invasive growth. Pseudohyphal growth was induced by supplementing YPD medium with 0.02% MMS or 20 mM $H_2O_2$ (Sigma-Aldrich) and incubating at 30°C with shaking at 200 rpm.

### Strain construction

The deletion of *PFY1* was carried out in *C. albicans* SC5314 using the *SAT*-flipper method as described previously (17). Briefly, the *SAT1*-flipper cassette flanked by 60 bp of upstream and downstream sequences of the *PFY1* gene was amplified by PCR. Then, the PCR products were transformed into SC5314 cells using the lithium acetate protocol. After transformation, the cells were recovered by culturing in fresh YPD medium at 30°C for 4 h with shaking at 200 rpm before spreading onto YPD-Nou plates. Two rounds of

**TABLE 1** *C. albicans* strains used in this study

| Strain | Relevant genotype | Source |
|---|---|---|
| SC5314 | Wild-type, clinical isolate | (16) |
| *pfy1Δ/Δ* | SC5314 *pfy1Δ::FRT/pfy1Δ::FRT* | This study |
| *pfy1Δ/Δ+PFY1* | SC5314 *pfy1Δ/pfy1Δ, RP10:: PFY1p-PFY1-FRT* | This study |
| *PFY1-GFP* | SC5314 *pfy1Δ/PFY1-GFP-FRT* | This study |
| Pfy1-GFP Act1-mCherry | SC5314 *pfy1Δ/PFY1-GFP, ACT1/ACT1-mCherry-FRT* | This study |

the transformation were required to obtained homozygous deletion mutants. Genomic DNA and total RNA were isolated from selected transformants to verify the mutations by PCR and RT-PCR analysis.

We cloned one copy of wild-type (WT) *PFY1* into the *Xho*I-*Hind*III sites of *pAG6* (16), linearized the plasmid with *Stu*I, and transformed it into the *pfy1Δ/Δ* mutant to obtain *PFY1* complemented strain.

To construct the *SAT1*-marked version of GFP or mCherry-tagging vectors, GFP or mCherry gene sequence, followed by the *CaURA3* terminator, was inserted into the *Apa*I–*Xho*I sites of *pSFS1*. To tag protein with GFP or mCherry at the C-terminus, the GFP or mCherry-*SAT1*-flipper cassette flanked by 60 bp of the coding sequence 5′ to the stop codon (without the stop codon) and 60 bp of the non-coding sequence 3′ to the stop codon was amplified by PCR. The PCR products were transformed into appropriate strains. Correct tagging was verified by PCR and Western blotting (WB) analysis. The oligonucleotide primers used to construct deletion cassette and fusion protein are shown in Table 2.

## Growth curves

Late-log phase *C. albicans* yeast cells were diluted to an $OD_{600}$ = 0.01 in 10 mL of YPD medium and were cultured at 30°C with shaking at 200 rpm. The culture (100 µL) was collected every 2 h, and the $OD_{600}$ was measured using a microplate reader (THERMO, Multiskan FC). The experiment was performed in triplicate.

## Susceptibility tests

*C. albicans* cells grown to the late-log phase in YPD medium were harvested and washed twice with sterile water. The cell suspensions were 10-fold serially diluted to generate suspensions containing $10^3$ to $10^6$ cells/mL, and 5 µL of each dilution was spotted onto YPD plates containing the indicated concentrations of MMS or $H_2O_2$ (Sigma-Aldrich). Growth was assessed by incubating the plates at 30°C for the indicated time. All experiments were performed at least three times.

**TABLE 2** Primers used in this study

| Name | Sequence (5′ to 3′) | Description |
|---|---|---|
| PFY1-A | TTTGTTCAATCTTTGGATTAAATATACAACAGATCTATTAAATACAATTGCAGAATCATTGCTGGGTACCGGGCCCCCCTCGAG | To delete *PFY1* |
| PFY1-B | CTTAACATCAACCATTAAATCTGGACAGGTATTGCACGTCCATGGATGCAATGAAAATATGGGCGAATTGGAGCTCCACCGCGG | To delete *PFY1* and tag *PFY1* with *GFP* |
| PFY1-GFP-A | CCAGGTGAAGCTACCACTCTTGTTGAAAAATTAGCCGATTACTTGATCAATGTCGGTTATGGGCCCATGTCTAAAGGTGAAGAA | To tag *PFY1* with *GFP* |
| Primer 01 | TTTCCAGTGTTAAGCAACACCTGG | |
| Primer 02 | TATAACTGTGCTAAAAGCCACGTA | To verify the mutations by PCR |
| Primer 03 | ATAACCGACATTGATCAAGTAATC | |
| ACT1-mCherry-A | TGGATTTCAAAACAAGAATACGACGAATCTGGTCCATCCATTGTTCACCACAAATGTTTCATGGTTTCAAAAGGTGAAGAAGAT | To tag *ACT1* with *mCherry* |
| ACT1-mCherry-B | AACAAAAAGAAGAATAACAAGAATACAAAACCAGATTTCCAGATTTCCAGAATTTCACTCGGGCGAATTGGAGCTCCACCGCGG | |
| RePFY1-A (Xho I) | CCGCTCGAGCCAGCTGAAAATTGTGCCAGTGAT | To clone *PFY1-GFP* into the *Xho*I-*Hind*III sites of *pAG6* |
| RePFY1-B (Hind III) | CCCAAGCTTTTATTTGTAC AATTCATCCA TACC | |
| CHS1-A | AAAAGTGTTGACCAGAACCGAG | |
| CHS1-B | ATGGCGTGAGCACAAATGA | |
| CHS2-A | TGATTTGGCAGCGATTAGTTAT | |
| CHS2-B | TCTTGTTGTGGAGGAGGTTCTT | qRT-PCR analysis |
| CHS3-A | GCTTGTAAGACTGTTGTCCCCG | |
| CHS3-B | AAATAGTAAATGTAATGGCTGCTGG | |
| CHS8-A | ATGGATGATGGTTCTCTTGTTG | |
| CHS8-B | GAATGTCTCTTCTTGATGGTGG | |

## Fluorescence microscopy

Log-phase *C. albicans* yeast cells were stained with 20 µg/mL Calcofluor White (Sigma-Aldrich) to visualize cell wall chitin. The cells were examined by differential interference contrast (DIC) and fluorescence microscopy (Olympus IX73). To visualize the chitin content, the exposure time for the images of Calcofluor White fluorescence was fixed.

Log-phase *C. albicans* yeast cells were stained with 1 µL, 200 T/mL Fluorescein Phalloidin (LABLEAD, G0059) to visualize actin. The cells were examined by DIC and fluorescence microscopy (Olympus IX73). To visualize actin organization, the exposure time for the images of Phalloidin fluorescence was fixed.

## Co-immunoprecipitation (Co-IP) and Western blotting

Co-IP and WB were performed as previously described by Han et al. (18). Briefly, mid-log phase C. *albicans* yeast cells were suspended in lysis buffer as previously described by Han et al. (18), and then mechanically lysed by five rounds of bead-beating at 5,500 rpm for 60 s at 4℃ by using Tomy Microsmash (Tomy-Seiko, MS-100R). Cell lysates were centrifuged, and the supernatant was transferred to a new tube to mix with 20–30 µL antibody-conjugated beads (Santa Cruz) that had been pretreated. After incubating at 4℃ for 8 h, the beads were washed for three times and then boiled for 5 min with the loading buffer added.

Proteins were resolved by SDS-PAGE and then transferred to a polyvinylidenedifluoride membrane (Millipore). The membrane was immersed in 5% skim milk dissolved in Tris-buffered saline containing 0.1% Tween 20 (TBST) for 1 h and then incubated in TBST containing the primary antibody at 4℃ for 8 h. After that, the membrane was washed for three times with TBST and then incubated with the secondary antibody conjugated with hydrogen peroxidase (Beyotime). Protein bands were detected by using the chemiluminescence system (Bio-Rad, ChemiDoc XRS+). mCherry and GFP antibodies were purchased from Beyotime (China).

## Quantitative RT-PCR

Total RNA was purified using the RNeasy Minikit and DNase treated at room temperature for 15 min using the RNase-free DNase set (Qiagen). cDNA was synthesized using the Maxima H Minus cDNA synthesis master mix (Thermo Scientific), and qRT-PCR was performed using the iQ SYBR green supermix (Bio-Rad) in 96-well plates.

## XTT reduction assay

This method was developed by Ramage et al. (19) and was used to quantify biofilm formation. *C. albicans* cells were grown to mid-log phase in YPD medium, washed twice with sterile phosphate-buffered saline (PBS), and suspended in Dulbecco's modified Eagle's medium (DMEM) medium at a density of $10^6$ cells/mL. Cell suspensions (100 µL) were added into wells of 96-well plates and incubated at 37℃, 5% $CO_2$, 100% humidity for 2 d. Then, non-adherent cells were removed by washing thrice with sterile PBS and leaving behind mature biofilm. A Colorimetric XTT Reduction Assay Kit (KeyGEN BioTECH) was used to quantify the biofilm formation. XTT-menadione solution (200 µL) was added to the prewashed biofilms and background control wells, and incubated at 37℃ in the dark for 3 h. Supernatants were transferred into new plates. Average values of XTT reduction reading at 450 nm of each strain are expressed as a percentage of the value of the WT strain.

## Murine model of disseminated candidiasis

The murine models were constructed as previously described by Han et al. (16). Mid-log phase *C. albicans* yeast cells were washed twice and diluted to $5 \times 10^6$ cells/mL with PBS. Thirteen female BALB/c mice per strain were injected via the tail vein with 200 µL of the cell suspension. The mice were monitored twice daily for survival for 21 d. To determine

the organ fungal burden, three mice were infected with each strain as described above and sacrificed at 48 h after the injection to surgically remove the kidney. One kidney from each mouse was removed, weighed, and homogenized. The homogenate was serially diluted in PBS and spread onto YPD plates for counting colony forming units (CFUs) per gram of kidney. Another kidney was fixed with formaldehyde, followed by 70% ethanol and then embedded in paraffin. Thin sections were cut and stained with periodic acid-Schiff staining for microscopic examination.

## Statistical analyses

All data shown in this study were from at least three independent experiments as means ± SD. The results of the *in vitro* experiments were analyzed with unpaired two-tailed Student's *t*-test. The results of survival curves and fungal burdens were analyzed using Kaplan-Meier test and Mann-Whitney test, respectively.

## RESULTS

### Profilin has a role in polarized growth in *C. albicans*

To explore the role of profilin in *C. albicans*, we generated *pfy1* mutant (*pfy1Δ/Δ*, *orf19.5076Δ/Δ*) in the SC5314 background (Fig. S1 and S2). Given that the *PFY1* homolog in *S. cerevisiae* is an essential gene, our initial assessment focused on whether *PFY1* affects the growth of *C. albicans*. In YPD liquid medium, *pfy1Δ/Δ* displayed a growth curve like that of the WT strain (Fig. 1A). On YPD solid plate, *pfy1Δ/Δ* showed no growth defects or temperature sensitivity when grown at 30°C/37°C (data not shown), indicating *PFY1* was not crucial for cell growth in *C. albicans*.

However, microscopic examination revealed that the *pfy1Δ/Δ* mutant exhibited a rounder and larger cell morphology compared to the WT strain (Fig. 1B and C). The aberrant cell morphology of *pfy1Δ/Δ* was restored by introducing one copy of wild-type *PFY1* at *RP10* locus (*pfy1Δ/Δ+PFY1*) (Fig. 1B and C). In addition, cellular polarity was investigated by staining cells with Calcofluor white (CFW) to monitor chitin distribution (20). In contrast to the WT and *pfy1Δ/Δ+PFY1* strains, where chitin staining was primarily localized to bud sites, *pfy1Δ/Δ* cells displayed a rounded shape with delocalized CFW staining patterns (Fig. 1B).

Profilin plays a critical role in actin cable assembly in *S. cerevisiae* (21). We next employed phalloidin to visualize actin filament distribution. In the WT cells, actin cables were visible in mother cells and oriented toward the small buds, where the cortical patches were concentrated; however, actin cables were not readily detected, and patches were dispersed throughout the mother cells in *pfy1Δ/Δ* cells, even when small buds were present (Fig. 1D). All these results suggest that profilin has a role in polarized growth of *C. albicans*.

### The loss of *PFY1* results in defective filamentous growth in *C. albicans*

As filamentous growth represented the most extreme form of polarized growth (20, 22, 23), the hyphal development of *pfy1Δ/Δ* under various inducing conditions was investigated. After induction in YPD containing 10% serum at 37°C for 1 h, both WT and *pfy1Δ/Δ* cells could initiate hyphal formation. However, after 3 h of induction period, *pfy1Δ/Δ* failed to produce hyphae or develop significantly shorter hyphae compared to the WT strain (Fig. 2A and B). On Spider plates, *pfy1* mutant exhibited a markedly reduced capacity to form wrinkly colonies, whereas the WT and reintegrant strains retained the ability for wrinkly colony formation (Fig. 2C). Similarly, *pfy1Δ/Δ* displayed a decreased invasion of agar (Fig. 2D). These results indicated that *PFY1* is required for hyphal growth.

To further investigate the role of *PFY1* in morphology transitions of *C. albicans*, late-log phase yeast cells were inoculated to fresh YPD medium with 0.02% MMS and 20 mM $H_2O_2$—agents known to cause DNA damage and promote pseudohyphal formation (24). As expected, upon induction with MMS and $H_2O_2$, WT and *pfy1Δ/Δ+PFY1* cells produced pseudohyphae; however, most *pfy1Δ/Δ* remained as yeast forms (Fig. 3A). Despite

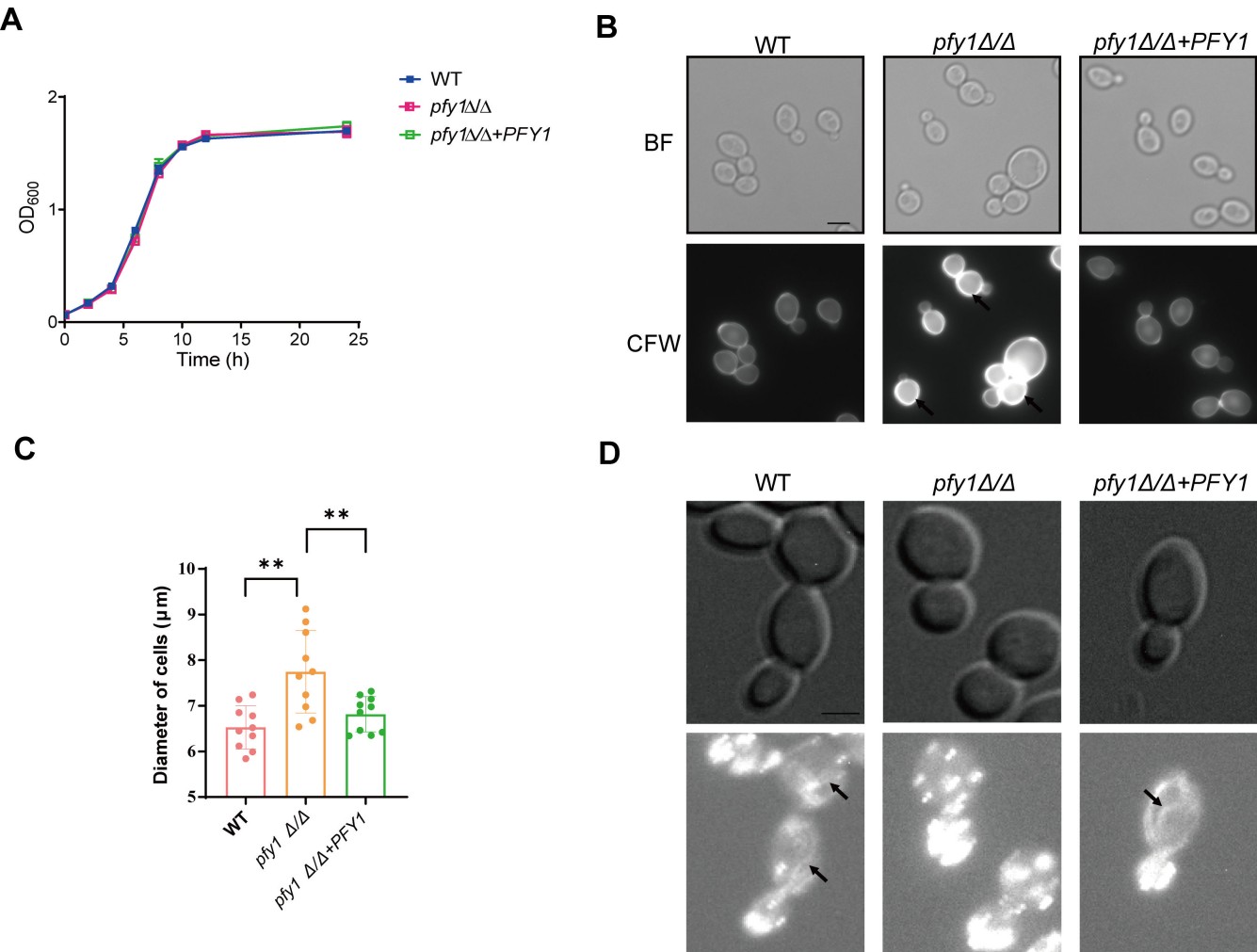

**FIG 1** *pfy1Δ/Δ* has normal growth and abnormal cell morphology. (A) Wild type (WT), *pfy1Δ/Δ*, and *pfy1Δ/Δ+PFY1* strains are depicted in a growth curve. (B) These same strains were visualized by bright field and fluorescence microscopy stained with calcofluor white (CFW). The arrows indicate delocalized chitin. Scale bar, 5 µm. (C) Statistical results of diameter of cells in (B). (D) Log-phase yeast cells (20 µL) of indicated strains were stained using rhodamine-phalloidin to visualize actin. The arrows indicate actin cables. The scale bar indicates 5 µm. The images shown in panels B and D are representative of three different experiments. Error bars in panel C are the standard deviations of three different data sets. Each individual data point was run in triplicate and averaged. Statistical significance between control strains and *pfy1Δ/Δ* strain was determined with Student's *t*-test (** represents $P < 0.01$).

demonstrating defects in genotoxic stress-induced filamentous growth, *pfy1* mutants did not exhibit altered sensitivity to either MMS or $H_2O_2$ (Fig. 3B).

## Profilin interacts with Act1 in *C. albicans*

Polarized growth is essential for cell morphogenesis and development in *C. albicans*, which is tightly regulated by assembly of actin (25). Profilin has been shown to modulate actin dynamics across various organisms, including fruit flies, yeast, and mammals (26). To determine whether Pfy1 interacts with Act1, *PFY1-GFP* was introduced into *pfy1Δ/Δ* under its own promoter. The expression of the fusion protein Pfy1-GFP was validated through Western blot analysis (Fig. S3). Furthermore, defects in hyphal development observed in *pfy1Δ/Δ* upon serum induction were rescued by the introduction of *PFY1* tagged with GFP (Fig. 2A and B). These results demonstrated that Pfy1-GFP was fully functional. Subsequently, we tagged Act1 at its C terminus with mCherry and expressed it in the strain containing Pfy1-GFP. Immunoprecipitation of Act1-mCherry was able to pull down GFP-tagged Pfy1 in the yeast or hyphal cells (Fig. 4). This interaction was

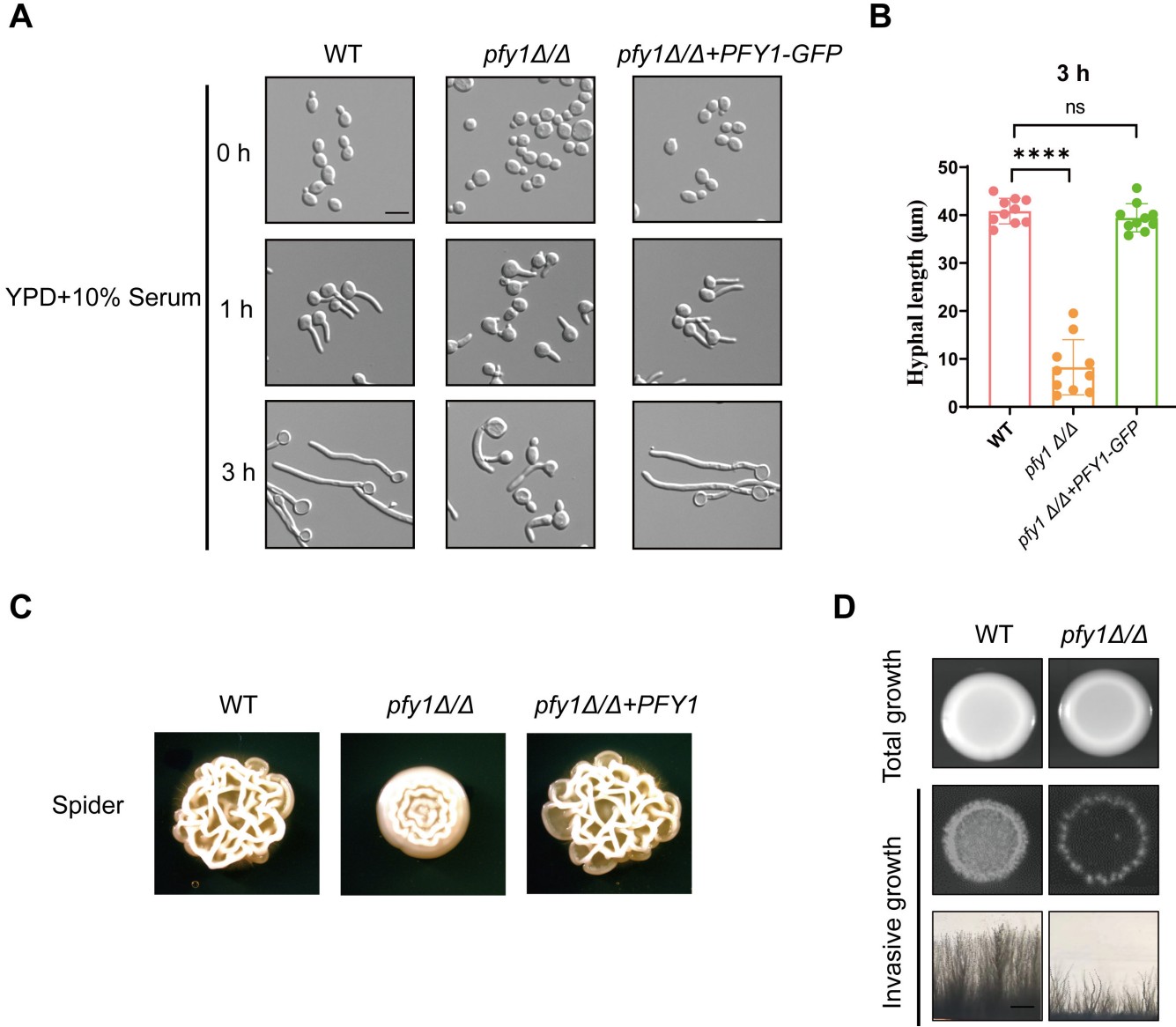

**FIG 2** The disruption of *PFY1* results in a defective filamentous growth in *C. albicans*. (A) WT, *pfy1Δ/Δ*, and *pfy1Δ/Δ+PFY1-GFP* strains were re-inoculated in YPD medium containing 10% serum and incubated at 37℃ for 3 h to take photos. Scale bars = 5 µm. (B) Statistical analysis of hyphal length in the same strains as described in (A) following a 3-h induction with 10% serum. (C) Colony morphology of WT, *pfy1Δ/Δ*, and *pfy1Δ/Δ+PFY1* strains grown on Spider plates at 30℃ for 7 d. (D) Invasive growth of WT and *pfy1Δ/Δ* strains. Strains were grown on YPD plates at 30℃ for 2 d. After this period, the plates were photographed before (total growth) and after (invasive growth) thoroughly washing the cells from the surface. Scale bars = 1 mm.

specific as it was not detectable in the control strain carrying only Pfy1-GFP (Fig. 4). Our findings indicate that Pfy1 could interact with Act1 independently of the yeast-hyphal transition.

## *C. albicans PFY1* impacts biofilm formation

Next, the functions of *PFY1* in biofilm formation were examined. We found that the density of biofilm decreased significantly in *pfy1Δ/Δ* cells, compared with WT and *pfy1Δ/Δ+PFY1* cells (Fig. 5A). Analyses with the 2,3-bis-(2-methoxy-4-nitro-5-sulfonyl)-2H-tetrazolium-5-carboxanilide salt (XTT) reduction assay revealed that the biofilm metabolic activity of *pfy1Δ/Δ* is reduced by ~75% compared to WT and *pfy1Δ/Δ+PFY1* strains

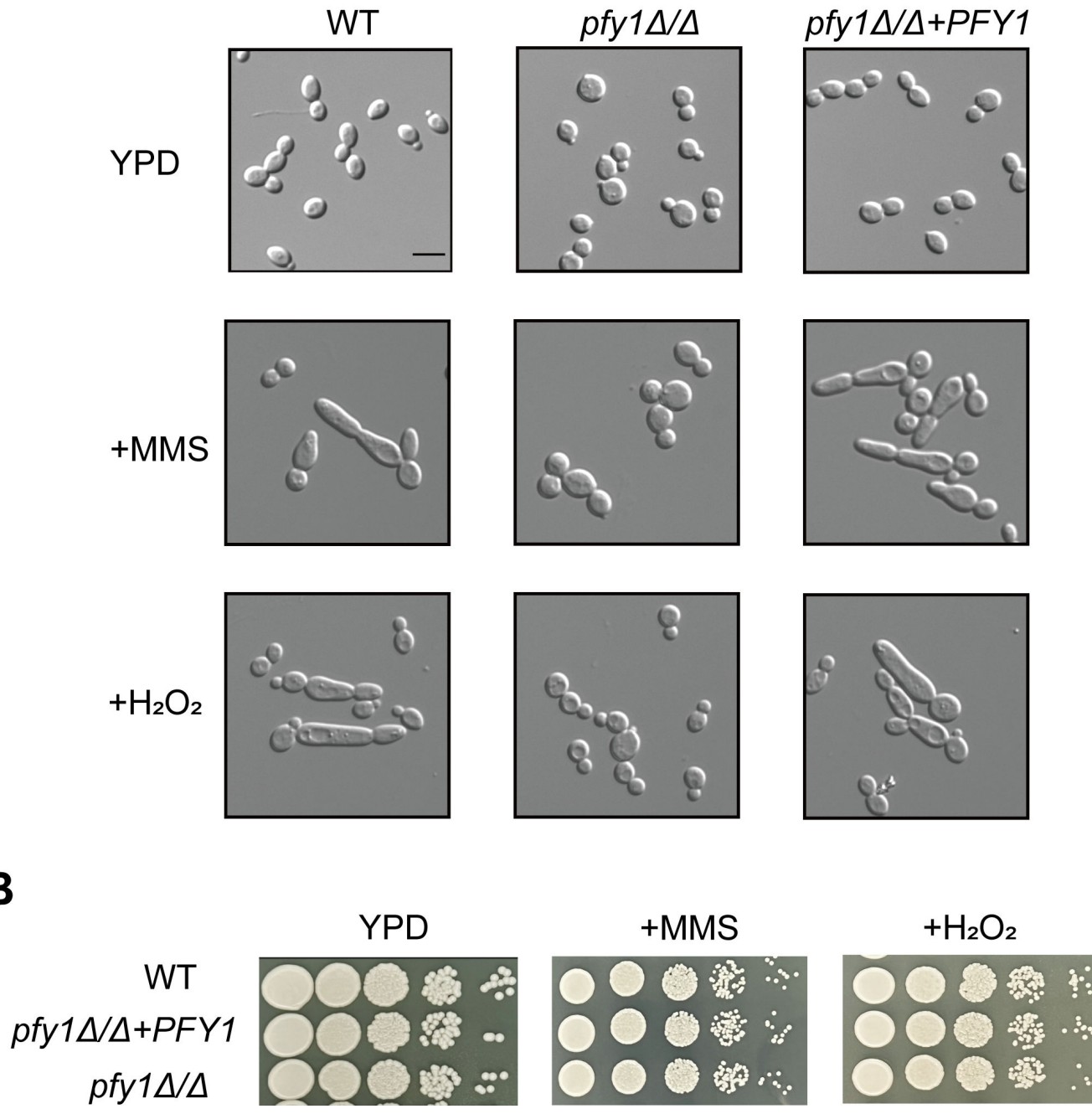

**FIG 3** *pfy1Δ/Δ* cells are defective in formation of pseudohyphae. (A) WT, *pfy1Δ/Δ*, and *pfy1Δ/Δ+PFY1* strains were re-inoculated in YPD medium containing 0.02% MMS or 20 mM $H_2O_2$ and incubated at 30°C for 6 h to take photos. Size bars = 5 µm. (B) Log-phase yeast cells of the same strains as described in (A) were serially diluted 10-fold and spotted onto YPD plates supplemented with 0.02% MMS or 20 mM $H_2O_2$. The YPD plates were incubated at 30°C for 48 h.

(Fig. 5B). These results demonstrated that lacking *PFY1* is defective in biofilm formation in *C. albicans*.

## *PFY1* contributes to cell wall stress responses

As shown in Figure 1B, staining with CFW revealed an increased fluorescent signal associated with the cell wall in *pfy1Δ/Δ* compared to WT and *pfy1Δ/Δ+PFY1* strains.

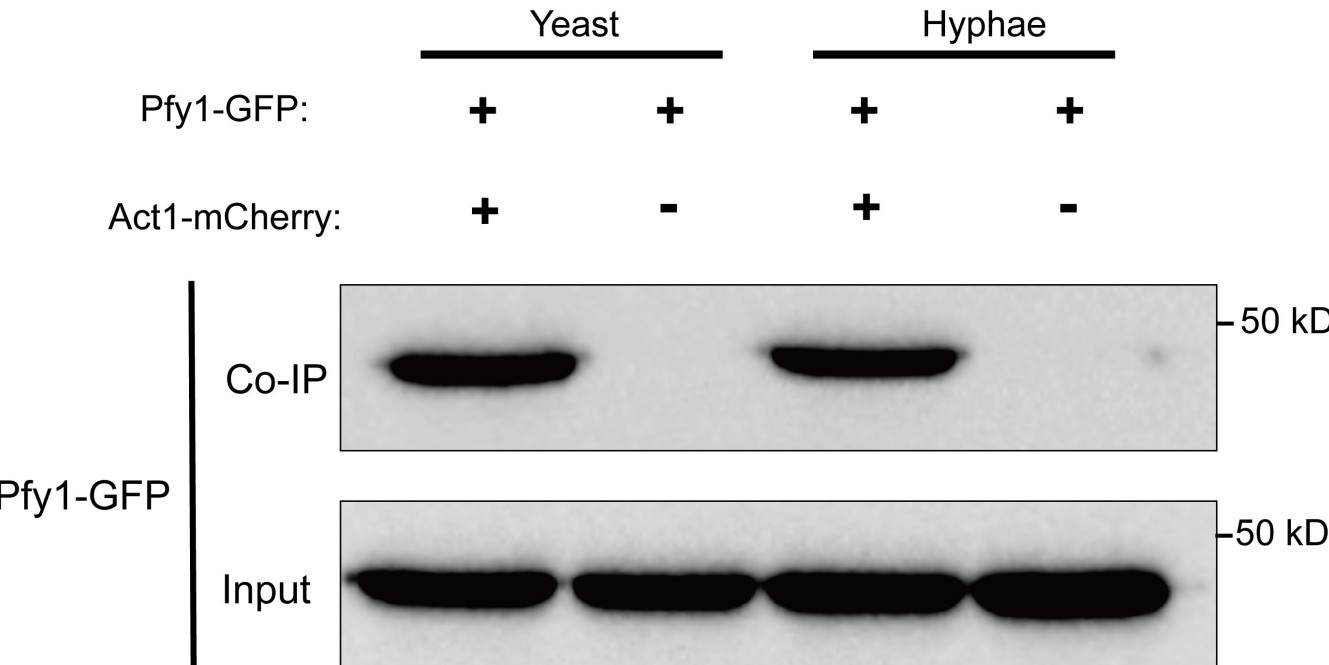

**FIG 4** Profilin interacts with actin. Co-IP of Pfy1 with Act1. Anti-mCherry conjugated beads were used for pull-down from log-phase yeast cells' and hyphae cells' lysates, and the pull-down products were probed with anti-GFP in WB analysis. As an input control, cell lysates were analyzed by Western blotting with the anti-GFP antibody.

Moreover, qRT-PCR analyses revealed that the expression of chitin synthase (*CHS1, CHS2, CHS3, CHS8*) was elevated in *pfy1Δ/Δ* (Fig. S4). This finding motivated us to further investigate cell wall integrity and function. The susceptibility of WT, *pfy1Δ/Δ*, and *pfy1Δ/Δ+PFY1* strains to a panel of cell wall stress agents (CFW, Congo Red, SDS) was assessed. Under these conditions, *pfy1Δ/Δ* had a dramatic growth defect compared to WT and *pfy1Δ/Δ+PFY1* strains (Fig. 6A). Additionally, transmission electron microscopy revealed notable differences in cell wall structure among these strains; specifically, the thickness of the cell wall in *pfy1Δ/Δ* was substantially greater than that in either WT or *pfy1Δ/Δ+PFY1* (Fig. 6B and C). These results suggested that Pfy1 was involved in remolding the cell wall in response to cell wall stressor.

Furthermore, the susceptibility of *pfy1Δ/Δ* to caspofungin, a non-competitive inhibitor of β-1,3-glucan synthase, was evaluated. *C. albicans pfy1Δ/Δ* displayed a substantial growth defect under caspofungin treatment (64/128 ng/mL), while the growth defect observed for WT and *pfy1Δ/Δ+PFY1* strains was minor (Fig. 6D).

## The loss of *PFY1* attenuates the virulence of *C. albicans* in the systemic candidiasis mouse model

To evaluate the role of Pfy1 in *C. albicans* pathogenicity, the virulence of the *pfy1Δ/Δ* was tested in a murine model of disseminated candidiasis. Over a 21-d observation period (Fig. 7A), 3 of 10 BALB/c mice infected by intravenous tail vein injection with *pfy1Δ/Δ* survived. In contrast, all the mice infected with WT or *pfy1Δ/Δ+PFY1* died, and the median survival times were 4–5 d, respectively. Fungal burden was also measured 2 d post-infection (Fig. 7B). The fungal burdens in the kidney of mice inoculated with *pfy1Δ/Δ* were significantly lower than those in mice infected with either WT or *pfy1Δ/Δ+PFY1*. Moreover, shorter hyphae and milder inflammation were observed in PAS (periodic acid-Schiff)-stained kidney sections from *pfy1Δ/Δ*-infected mice compared to those from WT and *pfy1Δ/Δ+PFY1*-infected mice (Fig. 7C). In summary, Pfy1 contributes to the colonization and invasion of *C. albicans* in mice.

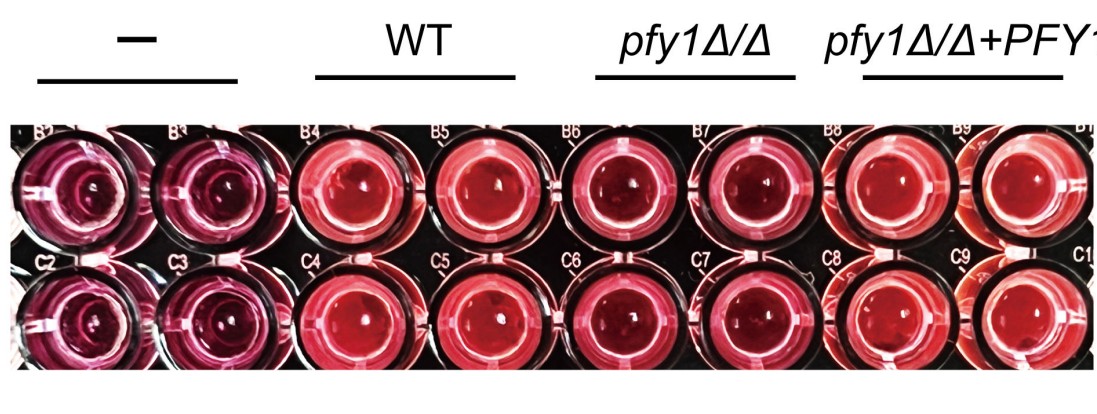

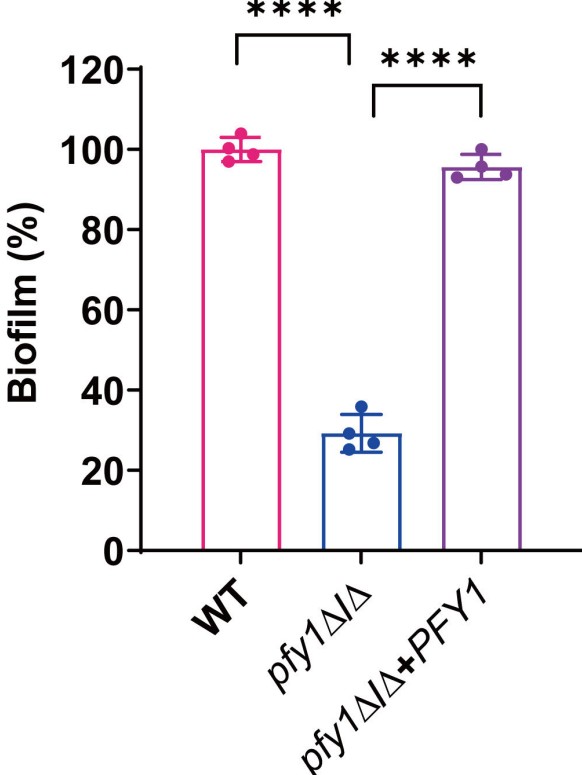

FIG 5 *pfy1Δ/Δ* forms an aberrant biofilm. (A) WT, *pfy1Δ/Δ*, and *pfy1Δ/Δ+PFY1* strains were re-inoculated in YPD medium and incubated at 30°C for 72 h. The biofilms formed were determined through XTT reduction assay. (B) Quantitative measurement of the biofilm formation by a colorimetric change resulting from XTT reduction was measured using a microtiter reader at 450 nm. **** represents $P < 0.0001$.

## DISCUSSION

The actin cytoskeleton serves as a major cellular component that facilitates a plethora of essential functions (27). The dynamic status of actin is modulated by accessory proteins that promote the rapid assembly and disassembly of filaments. In this study, we demonstrated that profilin, a key regulator of actin polymerization, was crucial for cell morphology, cell wall integrity, and virulence in *C. albicans*. Another essential regulator of actin cytoskeletal dynamic is cofilin, which is involved in selective sorting, environmental

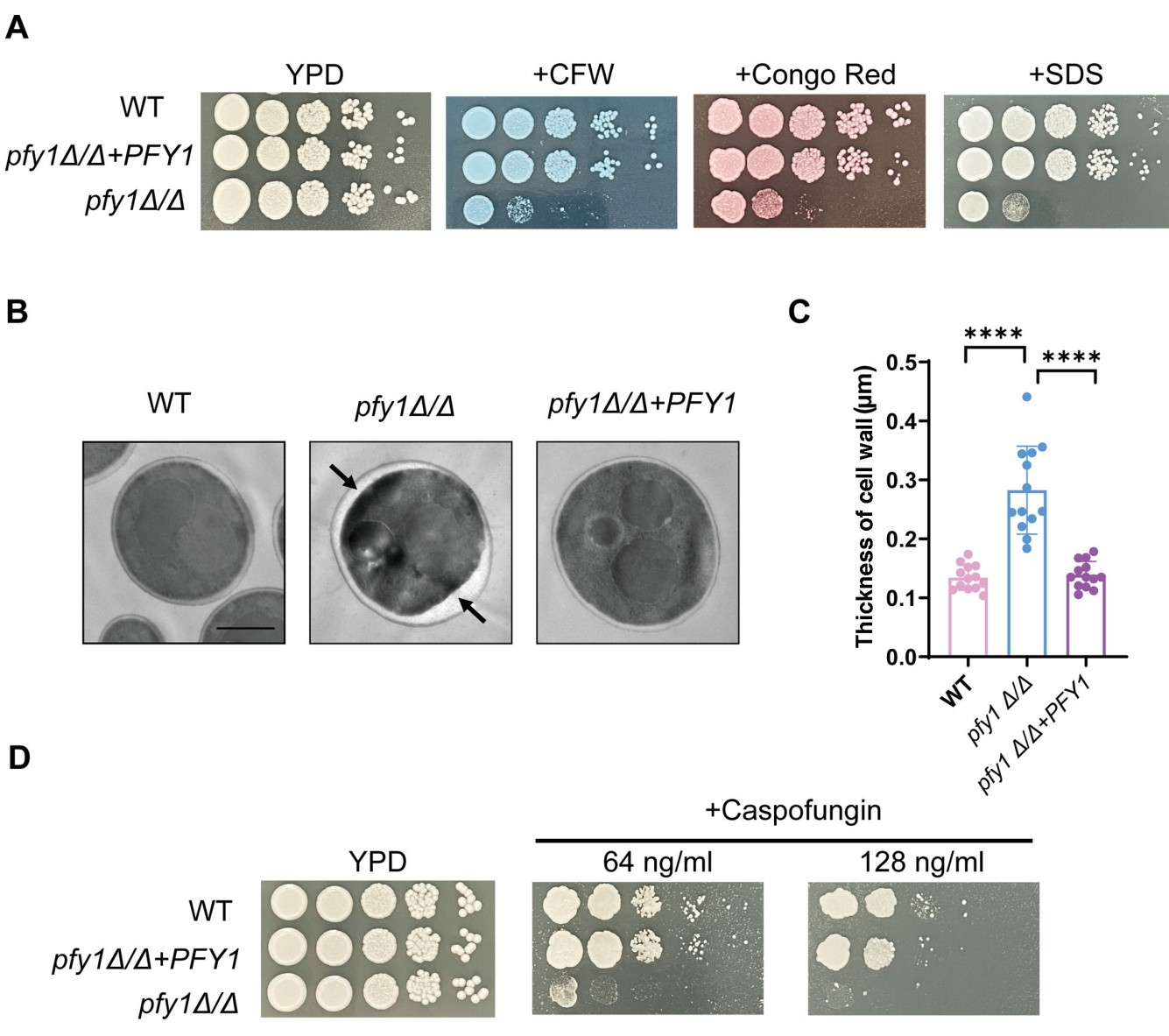

**FIG 6** *pfy1Δ/Δ* cells have thicker cell wall and are hyper-sensitive to cell wall stress. (A) WT, *pfy1Δ/Δ*, and *pfy1Δ/Δ+PFY1* strains were re-inoculated in YPD medium containing 80 µg/mL CFW and 100 µg/mL Congo Red or 0.02% SDS, and incubated at 30°C for 48 h. (B, C) Photos of cell wall thickness from the same strains were taken and analyzed. The black arrows show the cell wall thickness. Size bars = 2 µm. **** represents *P* < 0.0001. (D) These same strains were serially diluted 10-fold and spotted onto YPD plates supplemented with 64 or 128 ng/mL Caspofungin. The plates were incubated at 30°C for 48 h.

stress responses (including sensitivity to azoles drugs), and mitochondrial morphology in *S. cerevisiae* (28, 29). These findings underscore the importance of actin cytoskeleton dynamics in determining the fate of budding yeast.

Profilin is a multi-ligand protein structurally conserved among eukaryotes (30, 31). In *S. cerevisiae*, it binds with actin to regulate the dynamic actin cytoskeleton (32, 33) and coordinates with *SC*. Bni1p, Bem1p, Rho1p, Cdc24p, and Cla4p to regulate Ca²⁺ homeostasis and bud formation (10). Additionally, it directly interacts with cyclase-associated protein (CAP, known in yeast as Srv2) to catalyze the nucleotide exchange on actin monomers (34). Our research revealed that the disruption of *PFY1* resulted in striking morphological defects in both yeast and hyphal forms. Most *pfy1Δ/Δ* yeast cells were round instead of exhibiting a typical ellipsoidal shape and were enlarged in size (Fig. 1B). Furthermore, the *pfy1Δ/Δ* mutant cells showed severe defects in hyphal growth under various induction conditions (Fig. 2).We also confirmed the interaction between Pfy1 and

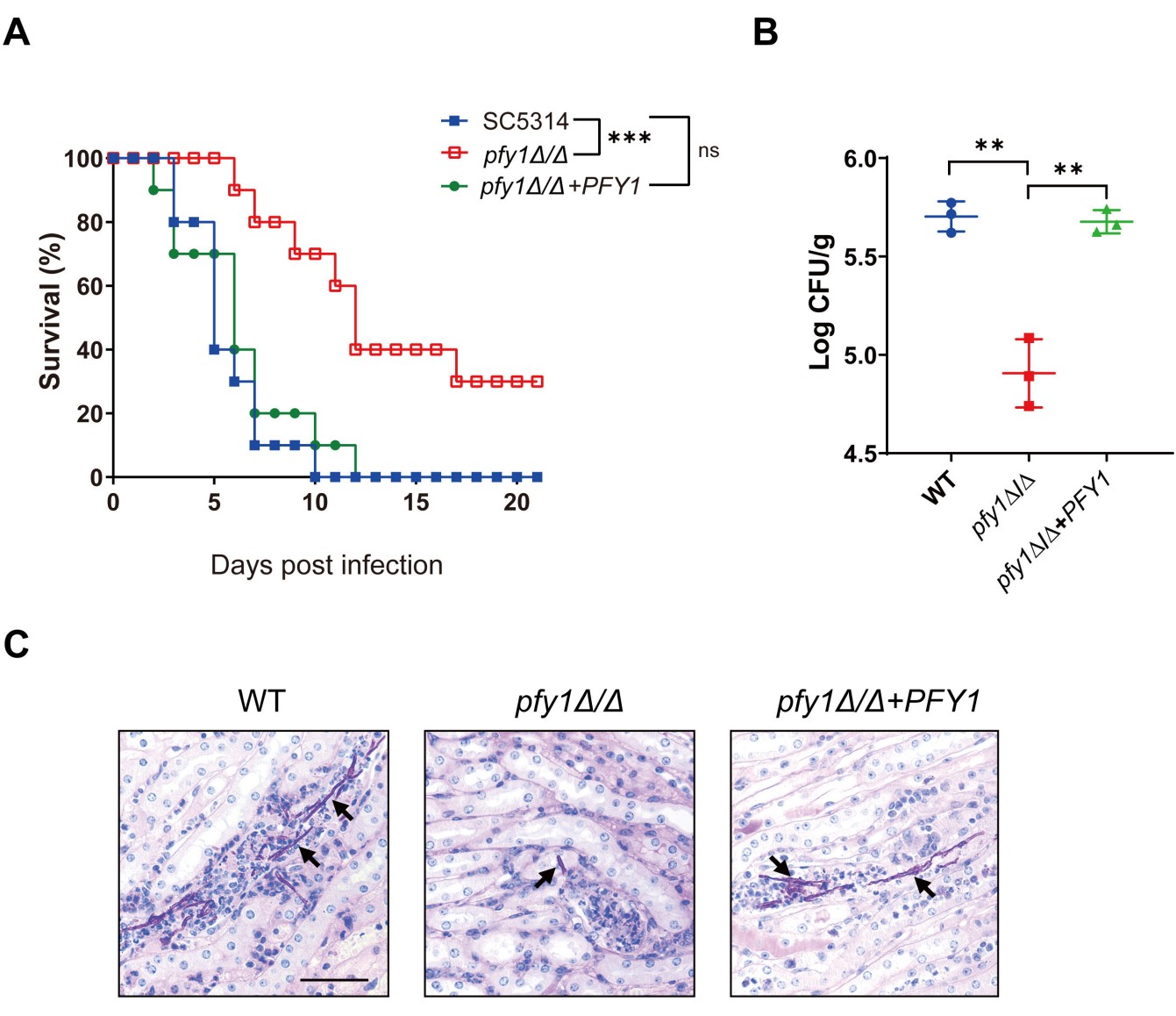

**FIG 7** *pfy1Δ/Δ* shows the impaired virulence in the systemic candidiasis mouse model. (A) BALB/c mice (*n* = 10) were injected via the tail vein with $10^6$ yeast cells of SC5314 (WT), *pfy1Δ/Δ*, *pfy1Δ/Δ+PFY1* strains, and monitored for survival over a period of 21 d. (B) Three mice were sacrificed after 48 h of the injection to determine the fungal loads in the kidney. (C) Conduct histological examinations of kidney sections. Size bars = 0.13 mm. Arrows indicate *C. albicans* cells in the renal tissues. Statistical analysis was performed using an unpaired two-tailed Student's *t*-test. ** represents *P* < 0.01; *** represents *P* < 0.001.

Act1 (Fig. 4), which localizes to polarized growth sites in budding and hyphal cells, indicating that profilin plays a role in polarized growth of *C. albicans* via binding with Act1. Future studies should investigate whether loss of *PFY1* could disturb the location of actin in *C. albicans*. In addition, profilin associated with polyproline helices that are present in a wide variety of proteins (35).

C. albicans can transform into different morphologies during commensalism and infection, such as yeast, hyphal, pseudohyphal, white, opaque, and gray cells (36). Among them, the regulation of yeast-hyphae transition and its influence on virulence have been studied widely. Here, we reported that deletion of *PFY1* in *C. albicans* resulted in abnormal morphogenesis characterized by enlarged cell size and impaired hyphal growth and reduced pathogenicity, suggesting that cell morphology is a key factor of virulence. The fungi switch between the different forms in response to external stimuli like pH, temperature, serum, nutrient starvation, etc. Apart from these, the cells also switch to alternate forms in response to certain genetic mutations or inhibition of cell

cycle progression. Our findings indicated that *Ca.* Pfy1 is not only involved in hyphal development but also essential for pseudohyphal formation in response to DNA damage agents MMS and $H_2O_2$ (Fig. 3A). These results highlighted the critical role of profilin in morphology transitions in *C. albicans*. Interestingly, we did not observe a defective growth of *pfy1Δ/Δ* upon $H_2O_2$ treatment, consistent with the notion that profilin function is dispensable for cell viability in response to oxidative stress in *S. cerevisiae* (37).

In *C. albicans*, the cell wall is composed of three primary layers: a basal chitin layer, followed by a central layer of β (1,3) and β (1,6)-glucan polymers, and an outer layer of mannosylated glycoproteins (38, 39). In agreement with previous studies demonstrating the impact of chitin content on cell wall integrity in *C. albicans* (40, 41), we also observed chitin deposition and thicker cell wall in *pfy1Δ/Δ* cells (Fig. 1B, 6B, and C), suggesting Pfy1 contributes to cell wall remodeling. Pfy1 has been predicted to be involved in carbohydrate transport 42, which may account for this phenomenon. It has been reported that overexpression of Pfy1 suppressors Mid2, Rom2, and Syp1 in *S. cerevisiae* also resulted in an abnormally thick cell; however, the thickness of the cell wall in *pfy1Δ* was similar to wild type in *S. cerevisiae* (21).

Both hyphal growth and cell wall integrity are the important virulence factors of *C. albicans*. In a systemic candidiasis mouse model, we observed significantly reduced mortality rates in mice infected with *pfy1Δ/Δ* strains and found *C. albicans* hyphae only in the kidneys of mice infected with either the *PFY1* complemented strain or the WT strain (Fig. 7A and B). Besides, *pfy1Δ/Δ* confers hypersensitivity of caspofungin (Fig. 6D), one of the first-line antifungals used to treat systemic candidiasis. These findings suggested that profilin could serve as a target for developing new antifungal drugs possibly for use in combination therapies with caspofungin for treating invasive candidiasis.

## ACKNOWLEDGMENTS

We thank Prof. Y. Lu (Wuhan University, China) for constructive suggestions.

This work was supported by research grants for Q.H.: National Natural Science Foundation of China (32100157), Guizhou Province Science and Technology Plan project (ZK [2023] generally 332) and Yichang Medical and Health project (A24-2-056).

## AUTHOR AFFILIATIONS

[1]School of Life Sciences, Beijing University of Chinese Medicine, Beijing, China
[2]The Third Clinical Medical College of the Three Gorges University, Gezhouba Central Hospital of Sinopharm, Yichang, Hubei, China
[3]Hubei Key Laboratory of Tumor Microenvironment and Immunotherapy, School of Basic Medicine, China Three Gorges University, Yichang, Hubei, China
[4]Yichang Key Laboratory of Infection and Inflammation, School of Basic Medicine, China Three Gorges University, Yichang, China
[5]Key Laboratory of Microbiology and Parasitology of Education Department of Guizhou, School of Basic Medical Science, Guizhou Medical University, Guizhou, China

## AUTHOR ORCIDs

Yueqing Wang  http://orcid.org/0000-0002-9507-6563
Qi Han  http://orcid.org/0000-0002-7944-9147

## AUTHOR CONTRIBUTIONS

Xun Sun, Data curation, Formal analysis, Methodology, Writing – original draft, Investigation | Yueqing Wang, Data curation, Formal analysis, Methodology, Writing – original draft, Investigation | Xiaomin Yang, Validation | Xi Xiang, Methodology | Lili Zou, Methodology | Xiaowen Liu, Methodology | Gang Luo, Methodology | Qi Han, Conceptualization, Writing – review and editing, Methodology, Resources, Supervision

## ADDITIONAL FILES

The following material is available online.

### Supplemental Material

**Supplemental figures (Spectrum02593-24-s0001.docx).** Fig. S1 to S4.

### Open Peer Review

**PEER REVIEW HISTORY (review-history.pdf).** An accounting of the reviewer comments and feedback.

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
