## [Reviewer comments · Microbiology Spectrum]

Microbiology Spectrum

Profilin Pfy1 is critical for cell wall integrity and virulence in *Candida albicans*

Xun Sun, Yueqing Wang, Xiaomin Yang, Xi Xiang, Lili ZOU, Xiaowen Liu, Gang Luo, and Qi Han

Corresponding Author(s): Qi Han, Beijing University of Chinese Medicine

Review Timeline:

Submission Date:	October 31, 2024
Editorial Decision:	November 24, 2024
Revision Received:	January 9, 2025
Accepted:	January 24, 2025

Editor: Chengshu Wang

Reviewer(s): The reviewers have opted to remain anonymous.

Transaction Report:

DOI: <https://doi.org/10.1128/spectrum.02593-24>

Re: Spectrum02593-24 (Profilin Pfy1 is critical for cell wall integrity and virulence in *Candida albicans*)

Dear Prof. Qi Han:

Thank you for the privilege of reviewing your work. Below you will find my comments, instructions from the Spectrum editorial office, and the reviewer comments.

Revision Guidelines

Sincerely,
Chengshu Wang
Editor
Microbiology Spectrum

Reviewer #1 (Comments for the Author):

The study investigates the role of profilin, an actin-binding protein, in *Candida albicans*, a major fungal pathogen. The authors present evidence suggesting that profilin plays a role in the morphogenesis, a process that is closely linked to the regulation of the actin cytoskeleton. Additionally, the *pfy1* mutant strain was found to be hypersensitive to cell wall stress and exhibited a thicker cell wall than wild-type cells, suggesting that profilin is crucial for maintaining cell wall integrity. The authors further demonstrate that profilin is important for the virulence of *C. albicans* in a murine model of systemic infection.

Overall, the findings provide insights into the cellular functions of profilin in *C. albicans*. My main questions and suggestions for the authors are:

1. The method used to determine actin filament distribution appears to lack rigor. In Figure 1D, where actin cables are stated to be absent, some patches are detected in the mother cells of *pfy1* mutant strain. The mislocalization of actin patches to the mother cell may hinder the observation of the less distinct actin cables present within the cell. Hence the statement that actin cables are absent (Lines 210) should be rephrased. Also, the authors should state the number of cells analyzed in this study.
2. It is recommended to investigate the agar invasion of the *pfy1* mutant to support the role of PFY1 in hyphal formation on solid medium (Spider agar plate). Additionally, Figure 2C is not referenced in the manuscript.
3. Statistical analysis is required to determine whether the differences observed in Figure 7A are statistically significant.

Reviewer #2 (Comments for the Author):

The manuscript "Profilin Pfy1 is critical for cell wall integrity and virulence in *Candida albicans*" is well written and supported by the facts and the data. However, the authors need some minor corrections in the manuscript as shown below:

Line 130: MMS is abbreviated but Line 230 shows the full form. check this. Same goes for H₂O₂

Line 343: "that overexpression SC" check this

Line 351: "one of the first-line antifungals used to treat systemic candidiasis, in vitro assays" check this. I couldn't understand this.

Check for the reference page as at some place et al. mentioned.

It's good to show a figurative representation of mutant created and add that in a figure.

Did the authors do any RNA studies on the mutant just to see the genes getting upregulated or downregulated? It would be good to include that.

Dear editor and reviewers,

We feel great thanks for your professional reviews work on our manuscript. As you are concerned, there are some flaws that need to be addressed. According to your suggestions, we have made extensive corrections to our previous draft (highlighted in red), the detailed corrections are listed now.

Best!

Qi Han

Reviewer #1 (Comments for the Author):

The study investigates the role of profilin, an actin-binding protein, in *Candida albicans*, a major fungal pathogen. The authors present evidence suggesting that profilin plays a role in the morphogenesis, a process that is closely linked to the regulation of the actin cytoskeleton. Additionally, the *pfy1* mutant strain was found to be hypersensitive to cell wall stress and exhibited a thicker cell wall than wild-type cells, suggesting that profilin is crucial for maintaining cell wall integrity. The authors further demonstrate that profilin is important for the virulence of *C. albicans* in a murine model of systemic infection.

Overall, the findings provide insights into the cellular functions of profilin in *C. albicans*. My main questions and suggestions for the authors are:

1. The method used to determine actin filament distribution appears to lack rigor. In Figure 1D, where actin cables are stated to be absent, some patches are detected in the mother cells of *pfy1* mutant strain. The mislocalization of actin patches to the mother cell may hinder the observation of the less distinct actin cables present within the cell. Hence the statement that actin cables are absent (Lines 210) should be rephrased. Also, the authors should state the number of cells analyzed in this study.

Thanks for your suggestions. We apologize for the inappropriate conclusion. The statement is now changed to “actin cables were not readily detected and patches were dispersed throughout the mother cells in *pfy1* Δ/Δ cells” in revision (line 219-220). The number of cells analyzed in this study is indicated in the legend for Figure 1.

2. It is recommended to investigate the agar invasion of the *pfy1* mutant to support the role of PFY1 in hyphal formation on solid medium (Spider agar plate). Additionally, Figure 2C is not referenced in the manuscript.

Thanks for your suggestions. As shown in Figure 2D, the *pfy1* mutant exhibited a defect in agar invasion, consistent with the conclusion that PFY1 is required for hyphal growth. In the revision, Figure 2C has been referenced (lines 230-232).

3. Statistical analysis is required to determine whether the differences observed in Figure 7A are statistically significant.

Thanks for your suggestions. We have conducted a statistical analysis of Figure 7A, which demonstrated that the survival rate of mice infected with the *pfy1* mutant was significantly higher compared to mice infected with the wild-type strains.

Reviewer #2 (Comments for the Author):

The manuscript "Profilin Pfy1 is critical for cell wall integrity and virulence in *Candida albicans*" is well written and supported by the facts and the data. However, the authors need some minor corrections in the manuscript as shown below:

Line 130: MMS is abbreviated but Line 230 shows the full form. check this. Same goes for H₂O₂

Thanks for your suggestions. We have corrected these issues in revision (line 132 236 237).

Line 343: "that overexpression SC" check this

Thanks for your suggestions. We have corrected it (line 349-350).

Line 351: "one of the first-line antifungals used to treat systemic candidiasis, in vitro assays" check this. I couldn't understand this.

Thanks for your kindly help. We have removed "in vitro assays" in revision (line 356-357).

Check for the reference page as at some place et al. mentioned.

Thanks for pointing this out. We have corrected this issue in revision.

It's good to show a figurative representation of mutant created and add that in a figure.

Thanks for your suggestions. We have added a schematic diagram of the construction of *pfy1* mutant in Figure S1.

Did the authors do any RNA studies on the mutant just to see the genes getting upregulated or downregulated? It would be good to include that.

Thanks for your suggestions. qRT-PCR analysis indicated that the expression of chitin synthases was elevated in *pfy1Δ/Δ*, which is consistent with the results obtained from CFW staining. And this data was shown in Figure S4.

Re: Spectrum02593-24R1 (Profilin Pfy1 is critical for cell wall integrity and virulence in *Candida albicans*)

Dear Prof. Qi Han:

Your manuscript has been accepted, and I am forwarding it to the ASM production staff for publication. Your paper will first be checked to make sure all elements meet the technical requirements. ASM staff will contact you if anything needs to be revised before copyediting and production can begin. Otherwise, you will be notified when your proofs are ready to be viewed.

Sincerely,
Chengshu Wang
Editor
Microbiology Spectrum

Reviewer #1 (Comments for the Author):

The authors have addressed my primary concerns with the manuscript.

Reviewer #2 (Comments for the Author):

Dear Authors,
Thank you for addressing and working on the comments. I am glad about your final revised version, however there are some minor corrections to be done that I have mentioned in the main text and supplementary files. Please address those.

1 **Profilin Pfy1 is critical for cell wall integrity and virulence**

2 **in *Candida albicans***

[revised manuscript text omitted]
	CCAGGTGAAGCTACCACTCTTGTTG AAAATTAGCCGATTACTTGATCAAT GTCGGTTATGGGCCCATGTCTAAAGG TGAAGAA	To tag PFY1 with GFP
Primer 01	TTTCCAGTGTTAAGCAACACCTGG	To verify the mutations by PCR
Primer 02	TATAACTGTGCTAAAAGCCACGTA	
Primer 03	ATAACCGACATTGATCAAGTAATC	
ACT1-mCherry-A	TGGATTTCAAACAAGAATACGACG AATCTGGTCCATCCATTGTTACCAC AAATGTTTCATGGTTTCAAAGGTG AAGAAGAT	To tag ACT1 with mCherry
ACT1-mCherry-B	AACAAAAGAAGAATAACAAGAATA CAAAACCAGATTTCCAGATTTCCAG AATTTCACTCGGGCGAATTGGAGCT CCACCGCGG	

RePFY1-A (Xho I)	CCGCTCGAGCCAGCTGAAAATTGTG CCAGTGAT	To clone PFY1-GFP into the XhoI - HindIII sites of pAG6
RePFY1-B (Hind III)	CCCAAGCTTTTATTTGTAC AATTCATCCA TACC	
CHS1-A	AAAAGTGTTGACCAGAACCGAG	qRT-PCR analysis
CHS1-B	ATGGCGTGAGCACAAATGA	
CHS2-A	TGATTTGGCAGCGATTAGTTAT	
CHS2-B	TCTTGTTGTGGAGGAGGTTCTT	
CHS3-A	GCTTGTAAGACTGTTGTCCCCG	
CHS3-B	AAATAGTAAATGTAATGGCTGCTGG	
CHS8-A	ATGGATGATGGTTCTCTTGTTG	
CHS8-B	GAATGTCTCTTCTTGATGGTGG	

Figure S1. Schematic diagram of the construction of *pfy1* mutant

Figure S2. Verification of correct construction of *pfy1Δ/Δ*, *pfy1Δ/Δ+PFY1* strains by PCR
 The PCR primers used are shown on the top. Genomic DNA was extracted from each strain and used as the template for PCR. (A: Primers 01 and 02 , B: Primers 01 and 03)

Figure S3. Protein level of Pfy1-GFP was analysis by Western Blot

Expression of the fusion protein was analysis by Western blot. Representative blots of three independent experiments are shown.

Figure S4. Expression levels of chitin synthase were increased in *pfy1* Δ/Δ

qRT-PCR analysis of expression levels of chitin synthase (*CHS1*, *CHS2*, *CHS3*, *CHS8*) in *pfy1* Δ/Δ and WT cells. Data shown as means \pm SD of three independent experiments. Statistical analysis was performed using an unpaired two-tailed Student's *t*-test. ** represents $p < 0.01$; *** represents $p < 0.001$.